# Structural and Functional Basis for Understanding the Biological Significance of P2X7 Receptor

**DOI:** 10.3390/ijms21228454

**Published:** 2020-11-10

**Authors:** María Ángeles Martínez-Cuesta, María Amparo Blanch-Ruiz, Raquel Ortega-Luna, Ainhoa Sánchez-López, Ángeles Álvarez

**Affiliations:** 1Departamento de Farmacología, Facultad de Medicina y Odontología, Universidad de Valencia, 46010 Valencia, Spain; Maria.a.blanch@uv.es (M.A.B.-R.); orlura@uv.es (R.O.-L.); itziar.sanchez@uv.es (A.S.-L.); 2CIBERehd, Valencia, Spain

**Keywords:** P2X7 receptor, allosteric modulations, human P2X7 receptor isoforms, channel membrane proteins, ATP, P2X7 receptor physiological role

## Abstract

The P2X7 receptor (P2X7R) possesses a unique structure associated to an as yet not fully understood mechanism of action that facilitates cell permeability to large ionic molecules through the receptor itself and/or nearby membrane proteins. High extracellular adenosine triphosphate (ATP) levels—inexistent in physiological conditions—are required for the receptor to be triggered and contribute to its role in cell damage signaling. The inconsistent data on its activation pathways and the few studies performed in natively expressed human P2X7R have led us to review the structure, activation pathways, and specific cellular location of P2X7R in order to analyze its biological relevance. The ATP-gated P2X7R is a homo-trimeric receptor channel that is occasionally hetero-trimeric and highly polymorphic, with at least nine human splice variants. It is localized predominantly in the cellular membrane and has a characteristic plasticity due to an extended C-termini, which confers it the capacity of interacting with membrane structural compounds and/or intracellular signaling messengers to mediate flexible transduction pathways. Diverse drugs and a few endogenous molecules have been highlighted as extracellular allosteric modulators of P2X7R. Therefore, studies in human cells that constitutively express P2X7R need to investigate the precise endogenous mediator located nearby the activation/modulation domains of the receptor. Such research could help us understand the possible physiological ATP-mediated P2X7R homeostasis signaling.

## 1. Introduction

The P2X7 receptor (P2X7R) is the last receptor subtype within the P2X family to have been characterized [1]. Currently, it is the focus of much research as a possible therapeutic target due to its contribution to an extensive number of disorders related to inflammation, immunity, and cell death. P2X7R, like the other members of its family, utilizes adenosine triphosphate (ATP) as a ligand, but interestingly it presents a unique structure which is associated to an unclear activation mechanism caused by high extracellular ATP levels [effective concentration 50 (EC50)]~500 µM) [2,3,4], which contrasts with the much lower activation onset conditions of the other six members of the P2X family (EC50~10 µM) [5,6,7]. In healthy tissues, ATP is almost exclusively present inside the cells (~mM), with only low concentrations of the molecule (~nM) existing extracellularly, whereas ATP is released from immune cells in response to a tissue insult reaching high concentrations (~100 µM) in the milieu [3,8]. Therefore, the extracellular ATP concentrations required for P2X7R activation are not present under physiological conditions and are indeed difficult to reach in severe damage process. In this context, some studies support the existence of other pharmacological ligands and/or allosteric modulators of this receptor.

Although P2X7R has an ambiguous mechanism of action, there is a general consensus that the receptor is consistently associated with inflammation [9,10,11,12]. In addition, it has more recently been related to thrombosis, fibrosis [13], tumor progression [11], and neuronal disorders [14]. Reviewing the reports published to date about P2X7R, there is little information regarding a possible role of this receptor in basal cell signaling in physiological functions. The aim of the present work is to analyze the unique chemical structure, genetic polymorphisms/variants, inconsistent activation pathways, and specific cell-site localization of the human P2X7R in order to understand its physiological significance beyond its already well-accepted pathophysiological involvement.

## 2. The Human P2X7 Receptor: Particular Structure and Genetic Polymorphisms

### 2.1. The Homo-Trimeric P2X7 Receptor Protein

P2X7R is an ATP-gated ion channel receptor that was initially identified as the cytolytic P2Z receptor due to its requirements of high concentrations of ATP to become activated [15,16]. However, the similitude of its sequence of amino acids with that of the other six P2XR led to its reclassification as P2X7R [1].

The human P2X7R gene is comprised of 13 exons and is localized in chromosome 12q24.31 [17]. Moreover, the gene encoding the P2X7R subunit has been reported in at least 55 species. However, the recombinant P2X7R subunit has been characterized, besides in humans, in six more mammalian species (macaque, dog, panda, mouse, rat and guinea pig). These species share more than 75% similarity with human P2X7R, while in non-mammalian species, such as the zebrafish, the similitude is reduced to below 40% [18]. A subunit of human recombinant P2X7R is a 595 amino acid-long protein that consists of a large extracellular loop containing two ATP-binding sites, two alpha-helical transmembrane domains, and intracellular N- and extended C-terminal tails (Figure 1a) [14,19]. A computerized structure model of the human P2X7R based on the crystal structure of the zebra fish P2X4R, the human P2X3R, and the panda P2X7R [20,21,22,23] led to the resolution of a three-dimensional structure due to the co-assembly of three P2X7R subunits in a homo-trimeric receptor (Figure 1b). It is well accepted that a functional P2X7R is made up of three subunits (Figure 1b), and, in contrast to other P2XRs, it rarely hetero-trimerizes with other members of the P2X family. In this context, it has been shown that P2X7R subunits can join to other P2X subunits—mainly to P2X4R and less so to P2X2R and P2X5R—suggesting that other P2X receptors contribute to the cellular responses typically attributed to P2X7R [7,24,25,26,27,28]. The tri-dimensional structure of the P2X7R subunits resembles the shape of a dolphin, in which the extracellular region represents the body (with head and fins) and the two transmembrane helices (TM1 and TM2) represent the tail (Figure 1c). The three ATP-binding pockets, which are supplied by the ATP sites of each pair of two adjacent monomers (Figure 1d), conform to the active P2X7R, which requires three ATP molecules for gating [22,29], and consequently open the low body domain pore, which is formed by the central TM of each one of the three pairs of TM domains. Interestingly, it has been suggested that the occupancy of two of the three ATP-binding pockets is sufficient to activate other P2XRs [30,31]. Additionally, almost juxtaposed with these ATP-binding pockets, another three drug-binding pockets (Figure 1d) are thought to accommodate different compounds with inhibitory and/or allosteric modulatory properties of the P2X7R [20], such as peptides [32,33,34], drugs [35], antibiotics [36], and plant derivatives [37].

Moreover, the intracellular N- and C-terminus of the P2X7R subunits play an important role in the biological functions of the receptor, but the detail and complete structure of these termini remains unsolved. In contrast to other P2X receptors, the P2X7R subunit has a larger C-terminus with an extra 200 amino acid residues (Figure 1a) [38] containing a cysteine-rich domain and a lipopolysaccharide-binding domain, which are unique domains amongst the P2X subunit members. This C-terminus has been associated with the modulation of signaling pathway activation and post-translational modifications [38]. The N-terminus (Figure 1a) has a specific site for protein kinase C that regulates the desensitization of P2XR [39] and is thought to regulate the flow of Ca^2+^ through the channel [40], as well as controlling the receptor gating and facilitating its activation [41]. Likewise, mutations of the residues at the N-termini have been related to changes in the permeability of the rat P2X7R to organic cations [42] or in the hypersensitivity of the human P2X7R induced by acute plasma membrane cholesterol depletion [43].

### 2.2. Polymorphisms and Transcriptional Regulation of P2X7 Receptor

Human P2X7R is highly polymorphic and more than 150 single-nucleotide polymorphisms (SNPs) have already been identified in the extracellular loop and in the C-terminal domain [12,19,44,45], leading to diverse unpredictable functional consequences. At the extracellular loop, the SNPs Gly-159-Arg and Glu-186-Lys result in a loss of P2X7R function. At the C-terminal, the SNP Iso-568-Asn impairs the translocation of P2X7R to the membrane [12], while the Gln-460-Arg variant’s effects remain unclear. There are data to support the notion that Gln-460-Arg SNP itself (homozygous) does not affect the receptor’s function, while in heterozygous mice it has been associated with a neuronal disorder [46]. In a human embryonic kidney cell line (HEK-293) Gln-460-Arg overexpression was shown to result in a reduction in the P2X7R-dependent ethidium uptake [47].

Interestingly, the His-155-Tyr (in the extracellular domain) and Ala-348-Thr (at TM2) SNPs have been associated with an increase in ATP-activated ion channel function and affect the formation of pores [47]. The Pro-451-Leu SNP at the C-terminal displays impaired pore formation of the P2X7R after ATP binding and exacerbates pain in mice [48].

Apart from these P2X7R polymorphisms, nine splice variants of the common P2X7R subunit (P2X7A) have been described and classified, from P2X7B to P2X7J. These splice variants lead to distinctive receptor behavior [12,49]. For instance, P2X7B can form functional channels but not pores, which is a hotly debated issue in terms of the established activation of the receptor [50]. P2X7H does not seem to assemble as a functional receptor [51], while P2X7J is capable of joining with P2X7A subunits to form a non-functional heterotrimeric-receptor with a cell-protective mechanism [52].

Among the P2X7R gene transcription factors, specific protein 1 (SP1) and hypoxia–inducible factor 1 alpha (HIF-1α) have been identified. SP1, a member of a damage-activated transcription factor family, binds to the CG-rich binding site of the P2X7R promoter in neuronal cell lines [53] and has been related to epileptic crises in animal models [54]. HIF-1α has been shown to upregulate P2X7R in a carcinoma cell line [55], data that are in line with the cytotoxic role of this receptor in apoptosis and cell death. Moreover, the fact that HIF-1α mediates P2X7R expression in astrocytes but not in neurons [56] suggests a cell-specific expression pattern of P2X7R.

## 3. Extracellular ATP Levels Are Critical to the Full Activation of the P2X7 Receptor

P2X7R is widely described as a cell surface receptor in which the ectodomain is exposed extracellularly. Since most ATP is located intracellularly, the ATP released into the extracellular space becomes a relevant signaling molecule [57,58,59]. Hence, high extracellular concentrations of ATP (EC50~50 µM to 2.5 mM) are required for P2X7R activation [1,20,60,61,62,63,64,65,66]; thus, the mechanisms by which ATP is released are closely associated with the function of the receptor.

Non-regulated ATP release takes place as a result of damaged or dying cells through unspecific mechanisms such as membrane disruption. However, controlled ATP release allows inflammatory and non-inflammatory cells to finetune their activation via autocrine/paracrine signaling [67] through specific cellular structures such as exocytotic granules; plasma membrane-derived microvesicles; membrane channels—connexins (connexin-43), pannexins (pannexin-1), calcium homeostasis modulator channels (CALHM), and P2X7R; and specific ATP-binding cassette (ABC) transporters [68,69,70,71]. For P2X7R activation, membrane channels such as connexins and pannexins play a critical role. Connexins form gap junctions between two adjacent cells but can also form unopposed hemichannels that allow small hydrophilic molecules such as ATP and ions to cross the cellular bilayer. It has been suggested that connexin-43 is co-localized with P2X7R in macrophages that mediate intercellular communication via gap junction formation mediated by extracellular ATP [70]. In addition, the existence of pannexin-1 (panx)-1 channels named the “pannexin-1/ATP/P2X7R axis” has also been related to P2X7R activation [72] (Figure 2). This axis has been localized in human inflammatory cells (macrophages [73] and monocytes [74]), in diverse human tissues (epithelial [75] and neuronal [76]), and in cancer cell lines [77]. In fact, pannexins are a three-member family of transmembrane proteins (panx 1–3) which cannot form gap junctions [78,79,80] but which facilitate the appearance of ionic currents (small molecules below 1–2KDa), whose properties resemble “undocked” gap junction hemi-channels [81]. Interestingly, panx-1 forms protein–protein associations with the P2X7R, whose activation by extracellular ATP opens a typical ion channel that is selective for small cations (including Ca^2+^) within milliseconds. This is followed, seconds later, by a prolonged opening or activation of a large pore permeable to molecules up to 900 Da (fluorescent dye uptake as ethidium and Yo-pro-1) [73,82], which contributes to cell death [83].

The final balance of extracellular ATP at a particular site of the cell, specifically surrounding the P2X7R, is not exclusively dependent on the amount of ATP released but also depends on the degree of ATP metabolism mediated by two cell-surface ecto-enzymes in a tightly regulated process [84]. Thus, the conversion of ATP into adenosine monophosphate (AMP) is catalyzed by CD39 (NTPDase-1) [85], whereas CD73 (ecto-5′-nucleotidase) catalyzes the dephosphorylation of AMP into adenosine [86] (Figure 2).

One of the obstacles for ATP’s action is its instant degradation. Therefore, the specific cell membrane co-localization of P2X7R near the site where ATP is released and at sufficient distance from degrading ATP enzymes could help to intensify a rapid endogenous extracellular ATP input that can trigger P2X7R.

## 4. Cell Localization of the P2X7 Receptor Is Crucial for the Nearby Interactions

P2X7R has been described mainly as a cell surface receptor. Uniform endogenous P2X7R expression has been demonstrated throughout the plasma membrane in various cell types, such as human lymphocytes [87] and HEK-293 cells (with the heterologous expression of rat P2X7R) [39]. In addition there is an asymmetric expression in intestinal epithelial cells, which move from an apical site in resting conditions to the basolateral surface during the phase of leukocyte transepithelial migration in vitro [88].

However, the precise P2X7R site in the plasma membrane in which ATP transport proteins and metabolism enzymes are located may not be the only issue to consider regarding the function of the receptor. In this context, P2X7R has been reported in specific places such as lipid rafts, apical and basolateral membranes, and even intracellular compartments. The localization of the receptor can be regulated by the potential modification of amino acid residues or domains within the C-terminus that could condition the presence of the P2X7R inside the cell or its movement to the surface [18].

Its presence in lipid rafts [89] has been related to the stimulation of lipid-signaling pathways (phospholipase A2, neuronal sphingomyelinase) [90]. Interestingly, regions within the N- and C-terminus close to the transmembrane domains contain cholesterol recognition amino acids, each of which contributes to cholesterol sensitivity to P2X7R. Impaired P2X7R activation and pore formation have been described as consequences of the increase in plasma membrane cholesterol in human and murine cells [43]. Additionally, heterologously expressed P2X7R seems to accumulate on the basolateral membranes in animal epithelial cells [91]. Finally, the P2X7R protein has been localized inside the cells [92]—intracellularly (leukocytes, platelets, neurons [93,94,95]), in the nuclei (epithelial and smooth muscle cells [96,97,98]), and in phagosomes (macrophages [99]). The specific role of intracellular P2X7R remains unclear, although some studies have pointed to the functions of differentiation and autophagy [93,100].

## 5. P2X7 Receptor Activation Theories and Transduction Pathways

The mechanism of action of P2X7R is still not well understood, thus unresolved theories need to be taken into consideration regarding the precise molecular pathways of P2X7R:

### 5.1. Is P2X7 Receptor a Regular Ligand-Gated Cation Channel?

When ATP is not present or exists in insufficient concentrations, P2X7R exists in a closed state (Figure 3a) and its conformation does not open its own ion channel (closed channel). In contrast, when there is sufficient concentration of ATP, the three molecules of ATP required for gating lead to conformational changes in the P2X7R by which it becomes fully activated (Figure 3b). However, P2X7R cannot be considered a regular ligand-gated ion channel, because, upon ATP stimulation, it is not permeable exclusively to small cations (with depolarization within milliseconds of the cell membrane with Ca^2+^ uptake or K^+^ efflux), whereas it seems to allow large cations and anionic molecules that normally do not go through ion channels to permeate (over 10 s to a minute). The P2X7R channel selectivity seems to differ depending on the experimental model, as it distinguishes between artificial systems which do not express the receptor endogenously, as shown by the majority of the studies performed, and cells that express the receptor naturally [101].

In systems in which P2X7R is artificially expressed, P2X7R activation mostly leads to cation-selective currents and is almost impermeable to anions. Thus, small cations such as Ca^2+^, K^+^, or Na^+^ can pass through the receptor channel and lead to the activation of the protein kinase B/serine and threonine kinase (PKB/AKT) pathway and NLR family pyrin domain containing (NRLP)3-inflammasome and a reduction in the electrochemical potential driving the K^+^ efflux, respectively [101]. To a lesser degree, large organic cations up to 130Da, such as N-methyl-D-glucamine (NMDG^+^), Tris(hidroximetil)aminometano (Tris^+^), and tetraethyl ammonium^+^, can also flow through the receptor channel, and despite their large size the uptake of cationic dyes (Yo-pro-1, ethidium bromide) seems to occur through the P2X7R itself [101]. Moreover, alterations in the ionic currents of these dye uptakes have been related to mutations of the C- and/or N-terminus, particularly the external deletion of the cysteine-rich region in the C-termini [1,102].

Nevertheless, in the few experiments undertaken in cells that constitutively express P2X7R, ATP stimulation seemed to activate a non-selective current, conducting both anions (Cl^−^, glutamate^−^, aspartate^−^) and cations (small and large) [101]. In particular, when added to macrophages, astrocytes, or macrophage-derived cell lines (J744, RAW264) exogenous ATP/2′(3′)-O-(4-Benzoylbenzoyl)adenosine 5′-triphosphate (BzATP,pharmacological P2X7R agonist) produces permeability, not only to cationic dyes such as Yo-pro-1 [73], but also to anionic dyes such as Lucifer yellow (433Da) and fluorescein (332Da) [103,104,105]. The anionic dye uptake by P2X7R stimulation seems to be cell-type dependent, since it occurs in macrophages but not in HEK293 cells. It should be remembered that P2X7Rs are not the only receptors expressed by native cells; thus, other purinergic receptors and ion channels also play a role in the induction of the response, thus making it difficult to assign a selective response to the P2X7R.

### 5.2. Does Permeation Occur through P2X7 Channel Receptor Itself or to Membrane Pore Proteins?

There are two permeation theories proposed to explain the passage of molecules which are usually too big to go through ion channels. These theories hold that activation gradually increases the permeability of the membrane through: (1) the enlargement of the receptor’s own ionic pore (Figure 3c) and (2) the activation of another “pore-forming” protein (Figure 3d). Electrophysiological studies have been somewhat inconsistent concerning the ion channel-to-pore enlargement theory, suggesting that P2X7R behavior is similar to that of other ion channels [106,107,108]; indeed, conductance studies have not shown any sign of dilatation during receptor activation (Figure 3c) [109]. Moreover, other studies in externally expressed P2X7R suggest that the cationic dye uptake takes place through the receptor itself [101]. On the other hand, although it has not been confirmed that proteins such as pannexins form a pathway that permeates large cation molecules [73] (Figure 3d) [110,111,112], a recent study supports that ATP-gated P2X7R requires chloride channels—which is a protein distinct to pannexins—to promote inflammation specifically in human—but not in murine—monocyte-derived macrophages [113]. The two permeability theories are not necessarily mutually exclusive, and additional studies specifically analyzing the membrane permeabilization sites near the receptor would help to better understand P2X7R function.

### 5.3. Does P2X7 Receptor Directly Trigger Intracellular Signaling Pathways?

P2X7R stimulation leads to the activation of diverse intracellular signaling molecules, including protein kinases PKB/AKT [114,115], phospholipases (phospholipase A2 [116], phospholipase D [117,118]), tyrosine phosphatase-β [119], epithelial membrane proteins (EMPs) [119,120], and the large molecular complex NLRP3 inflammasome [121,122].

For instance, P2X7R activation induces cyclooxygenase (COX) activation and the expression of extracellular signal-regulated kinase (ERK)/ mitogen-activated protein kinase (MAPK) kinases and phospholipase A2 [123,124] in human macrophages and fibroblasts. Studies in different cells—for example, in rat microglia—suggest that the P2X7R is coupled with c-Jun N-terminal kinase (JNK) and p38, but not ERK. This association is responsible for the activation of JNK and p38 via a protein tyrosine kinase-dependent mechanism that involves the release of tumor necrosis factor (TNF-α) [114]. In these cells, a putative tyrosine phosphorylation site within the P2X7R intracellular C-terminal domain (Y382-384) is involved in the P2X7R-induced release of TNF-α [125].

The NLRP3 inflammasome is a well-established pathway associated with K^+^ flux upon P2X7R activation [10,126,127]. A drop in the intracellular K^+^ causes NLRP3 inflammasome assembling and activation, probably via the cytoplasmic kinase never-in-mitosis A- related kinases (NEK), cleaving pro-caspase-1 into caspase-1 and causing the maturation of pro-inteleukin(IL)-1β and pro-IL-18 into IL-1β and IL-18, respectively [10] (Figure 4). These are not the only pro-inflammatory cytokines released by P2X7R activation; IL-6 and IL-1α are also released, but through an inflammasome-independent route [128,129]. The synthesis of immature pro-IL-1β requires the activation of the nuclear factor nuclear factor κB (NF-κB), and this factor seems to be involved in the cytokine gene expression promoted by the P2X7R [126] (Figure 4). In support of this, it has been proposed that the adaptor protein of the NF-κB (MyD88) interacts with the C-terminal domain of the P2X7R subunit [130] to provoke NF-κB activation. NLRPs other than NLRP3 are also implicated; in fact, the NLRP2 protein forms a complex with the P2X7R which is related with panx-1 and mediates IL-1β maturation and release [131,132]. The NRLP3/inflammasome route has been linked to pyroptosis via the activation of inflammatory caspases (casp), with differences occurring depending on the species (casp-1 and casp-11 in mice vs. casp-1, caps-2, and caps-5 in humans) [133]. The variability of the transduction pathways linked to P2X7R activation must be interpreted on the basis of the structural diversity of P2X7R domains surrounded by a singular molecular environment in each type of cell.

### 5.4. Can P2X7 Receptor Be Allosterically Modulated by Endogenous Molecules?

The high ATP concentrations (up to 2.5 mM) required for P2X7R activation imply important consequences, since concentrations over 1 mM lead to cell death or delayed cell death (pseudoapoptosis) depending on the duration of ATP exposure [112,114]. However, lower concentrations of ATP (<1 mM) can have the opposite effect, stimulating proliferation and prolonging cell survival [134]. In this context, a positive allosteric modulation of P2X7R that increases the sensitivity of this receptor to low extracellular ATP levels supports the relevance of this receptor in non-damaged tissue (Figure 5).

To date, an important number of negative or positive allosteric modulators have been described and we have complied and classified them as “endogenous” or pharmacological modulators in Table 1.

“Endogenous” allosteric modulators of P2X7R such as trace metals (divalent cations; Mg^2+^, Cu^2+^, Zn^2+^), play a negative modulator role, whereas phosphoinositides, key components of biological membranes, have a positive potentiation effect [135]. Copper or zinc modulation has been co-related with mutations of some extracellular amino acid residues (His and Asp) of the receptor [136,137], while inner leaflet membrane phosphoinositides have been shown to interact with the proximal C-terminal domain [138,139]. Studies in HEK293 cells expressing recombinant P2X7R which have been exposed to diverse lipids, including lysophosphatidylcholine and other lysolipids products of PLA2, have demonstrated an increase in the agonist potency of P2X7R, reinforcing the idea that membrane lipids can modulate P2X7R in vivo [140]. One study in a chinese hamster ovary (CHO) cell line suggested that cell surface glycosaminoglycans cause a positive modulation of the P2X7R [141]. Additionally, lipoglycans - such as lipopolysaccharide (LPS) - found in the outer membranes of bacteria could amplify receptor response by interacting with a LPS-binding domain in the extended C-terminal [142], supporting an increasingly clear link between the P2X7R and the immune response. Another endogenous messenger, nicotinamide adenine dinucleotide (NAD), a donor of adenosine diphosphate (ADP)-ribose moieties, is thought to act synergistically with ATP extracellularly to regulate P2X7R signaling in murine macrophages [143]. Interestingly, although NAD regulates the P2X7R in both macrophages and T cells, the mechanisms at play are different. While ADP-ribosylation is sufficient to open the P2X7R channel in T cells, P2X7R modification in macrophages does not gate the channel, but increases the ATP sensitivity [143]. Furthermore, data regarding alternative variants of the mouse P2X7R reveal that sensitivity to NAD is mediated through the variant P2X7K [144].

Many pharmacological drugs and natural products are also allosteric modulators of P2X7R. The anti-histamine clemastine was proposed as an extracellular binding positive allosteric modulator of the human recombinant P2X7R (transfected HEK-293) and natively expressed P2X7R (from human monocyte-derived macrophages and murine bone marrow macrophages) that sensitizes the receptor to lower ATP concentrations and facilitates its pore dilation [145]. Moreover, clemastine has been shown to augment the IL-1β release from LPS-primed macrophages [145]. The anti-inflammatory Tenidap, a non-fully characterized COX/5-lipoxygenase (LOX) inhibitor, enhances ATP-induced cytotoxicity mediated by P2X7R in mouse macrophages [35]. The anesthetics propofol and ketamine increase the P2X7R-mediated currents in a rat microglia cell line [146]. The antibiotic polymyxin B, which binds and neutralizes LPS, potentiates P2X7R, since it enhances ATP-mediated Ca^2+^ influx, membrane permeabilization, and cytotoxicity [36], perhaps through a mechanism related to an N-terminal fatty amino acid residue of the drug that has been shown to interact with the receptor’s extracellular domain [147]. Ginsenosides, a plant product, were shown to increase ATP-activated P2X7R-triggered cell death provoked by a non-lethal concentration of ATP in different macrophage cell lines [37,148]. The allosteric location of these ginsenosides may be in the P2X7R transmembrane domain [148], which is also shared by other P2X receptors, such as P2X4R [149]. Agelasine and garcinolic acid, other plant products, intensify the P2X7R responses in a human A375 melanoma cell line and in mouse microglial cells [150]. Ivermectin, a common P2X4R-positive allosteric modulator initially used to discriminate P2X4-from P2X7-mediated responses, was found to potentiate human but not murine-monocyte-derived macrophage P2X7R, suggesting the existence of a species selectivity for P2X7R [151]. Moreover, ivermectin is thought to drive P2X4R/P2X7R/pannexin-1 signaling in order to enhance cell death pathways in cancer cells [77]. In agreement with species-specific P2X7R, GW791343—an artificial compound—displays the opposite behavior towards P2X7R from HEK-293 cells, acting as a negative or positive modulator when these cells express the human or rat recombinant P2X7R, respectively [152].

## 6. Pathophysiological Role of P2X7 Receptor and Its Biological Significance

P2X7R is crucial to the inflammatory and/or immune response [12], since it is expressed in all immune cells (lymphocytes, macrophages, monocytes, neutrophils, basophils, dendritic cells, eosinophils, and mast cells) and participates in a large number of disorders. For instance, P2X7R is associated with autoimmune diseases (systemic lupus erythematosus, autoimmune hepatitis) [155] and contributes to the immune response in patients with viral infections (hepatitis C, human inmmunodeficiency virus-1, dengue-2 virus), with interesting dual anti-viral and pro-viral properties [156]. Inflammatory P2X7R involvement affects all organs and tissues, as this receptor has been implicated in thrombosis, asthma/allergic hyperactivity, Crohn’s disease/ulcerative colitis, glomerulonephritis, chronic dermatitis, and psoriasis [11,157,158,159,160,161]. Moreover, a sustained inflammatory response and other cell damage insults can lead to cellular degeneration, apoptosis, and necrosis. For example, the P2X7R is upregulated in various neuronal diseases and/or under stress stimuli, which can lead to excitotoxicity, neuroinflammation, and neuronal damage [162], while in the cardiovascular system it has been associated with myocardial injury and cardiac fibrosis after ischemia [13,163]. Furthermore, a role for the receptor in cell proliferation during wound healing has also been acknowledged in the activation of the downstream signaling that promotes cytoskeletal rearrangements and cell migration [164]. Additionally, P2X7R may be a key mediator of tumor invasion/metastasis by inducing proliferation or cell death depending on the tumor type [165,166], as reported in many human cancer studies (breast carcinoma, pancreatic duct adenocarcinoma, brain tumor, osteosarcoma, lung cancer, etc.) [11,167]. In fact, P2X7R has a dual role by which it causes cell proliferation (survival) versus cell death (cytolysis) depending on the concentration of ATP, which seems to condition the pathophysiological activity of this receptor [168]. Endogenous concentrations of ATP are related to the pro-cancerous basal activity of P2X7R, since the inhibition of this receptor in highly invasive breast and lung cancers has been shown to reduce cell migration and invasiveness [169,170]. In contrast, the prolonged full-activation of P2X7R induced by high exogenous concentrations of ATP results in anti-cancerous cytotoxic activity that may be therapeutic in certain cancers, such as leukemia [171]. In this context, P2X7R currently shows great potential as a therapeutic target in many fields of research [11,163].

Many of these processes, including inflammation, cell immunity, apoptosis, and necrosis, could be a result of the activation of the inflammasome/P2X7R complex and subsequent molecular patterns (cytokines and caspases) (Figure 4) in a high extracellular ATP environment, inexistent in normal conditions but required for the activation of the receptor. However, the existence of possible allosteric modulators that facilitate low concentrations of ATP—which are present in regular physiological conditions—to activate the receptor (Figure 5) shines light on a novel mechanism that could underlie the development of some of the diseases in which P2X7R is involved and has a bearing on its physio/pathological significance.

## 7. Conclusions

P2X7R is involved, together with its endogenous agonist ATP, in cell damage and/or death signaling in many species and cellular types. The mechanism responsible for these actions is not fully understood, but appears to be associated with cell permeability to large ionic molecules through the receptor itself or to nearby membrane proteins. Accordingly, most of the studies performed to date highlight the involvement of this receptor in processes such as inflammation, immune response, fibrosis, apoptosis, necrosis, neurodegeneration, and tumor progression. It is important for research to focus on the particular structure of this receptor—specifically, the extended C-terminal domain—together with the numerous polymorphisms detected in the receptor subunits, assembled in a homo- or hetero-trimeric receptor, and its precise cellular environment. Putting to one side the variability in the structure and activation of the receptor depending on the species and cell type tested, the results of past research may have been conditioned by the use of artificial, modified P2X7R, as most studies have been performed in an overexpressed system of recombinant murine and/or human P2X7R, while few have been carried out in human primary cells that endogenously express the receptor [113]. The consensus regarding the large extracellular ATP concentrations required for the receptor to be triggered—which, in turn, are associated with cell death—contrasts with the putative allosteric modulation of this receptor by endogenous mediators, which would convert the tiny regulated ATP inputs into activated functional P2X7R signals. However, these endogenous mediators need to be explored further, since it is possible that they can move from unresolved key/s component/s of the biological membranes and/or intracellular compartment to a particular extracellular endogenous molecule such as NAD. In addition, researchers should take into account the fact that the exogenous exposure of P2X7R to ATP, allosteric modulators, and/or drugs is not the same as when an endogenous mediator is released at a precise site near the activation/modulation domain of the receptor. Future studies in human tissues expressing native P2X7R focusing on endogenous ATP and/or allosteric modulators are likely to help us see beyond P2X7R cell damage to a possible role of physiological cell homeostasis signaling.

## Figures and Tables

**Figure 1 ijms-21-08454-f001:**
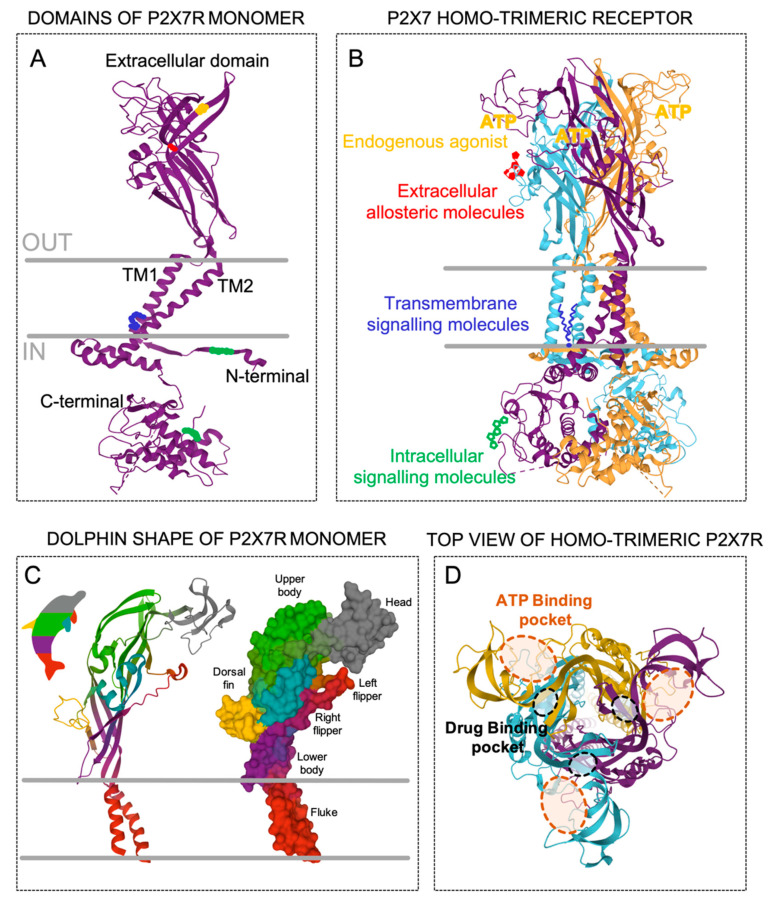
Topology of the P2X7 receptor (P2X7R). (**A**) Secondary structure of the P2X7R monomer with five domains; extracellular, transmembrane 1 (TM1), transmembrane 2 (TM2), intracellular N-termini, and intracellular C-termini. The colors into the domains represent possible specific molecular recognition sites. (**B**) Co-assembly of the three P2X7R monomers (purple, blue, and orange) position into the trimeric receptor showing possible surrounding molecules that could interact with the different domains, such as intracellular messengers at the N- and/or C- termini (green), membrane residues at the transmembrane domains (blue), and the endogenous agonist adenosine triphosphate (ATP, yellow) and/or other allosteric modulators (red) in the extracellular domain. (**C**) The P2X7 monomer represented as secondary structure (left) and molecular surface (right) styles resembling a dolphin. (**D**) Top view of the P2X7R structure embracing the three extracellular ATP-binding pockets (orange dashed lines) and the three additional drug-binding pockets (black dashed lines).

**Figure 2 ijms-21-08454-f002:**
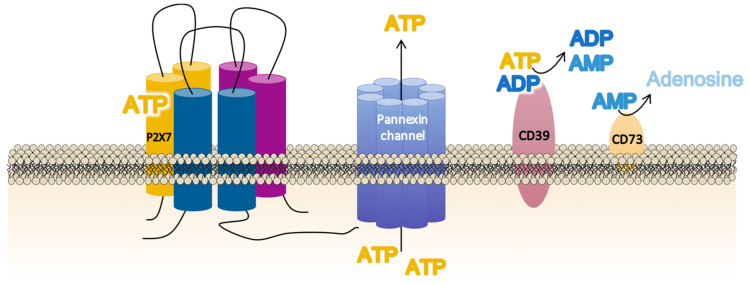
Schematic illustration of the main membrane proteins involved in the regulation of the extracellular adenosine triphosphate (ATP) levels required for P2X7 receptor (P2X7R) function. The P2X7R activation requires large ATP release through nearby membrane structures such as the pannexin channel, but the precise balance of the extracellular ATP concentrations is also dependent on the activity and membrane localization of the enzymes CD39 (NTPDase-1), which dephosphorylates ATP to adenosine diphosphate (ADP) and adenosine monophosphate (AMP), and CD73 (ecto-5′ecto nucleotidase), which turns AMP into adenosine.

**Figure 3 ijms-21-08454-f003:**
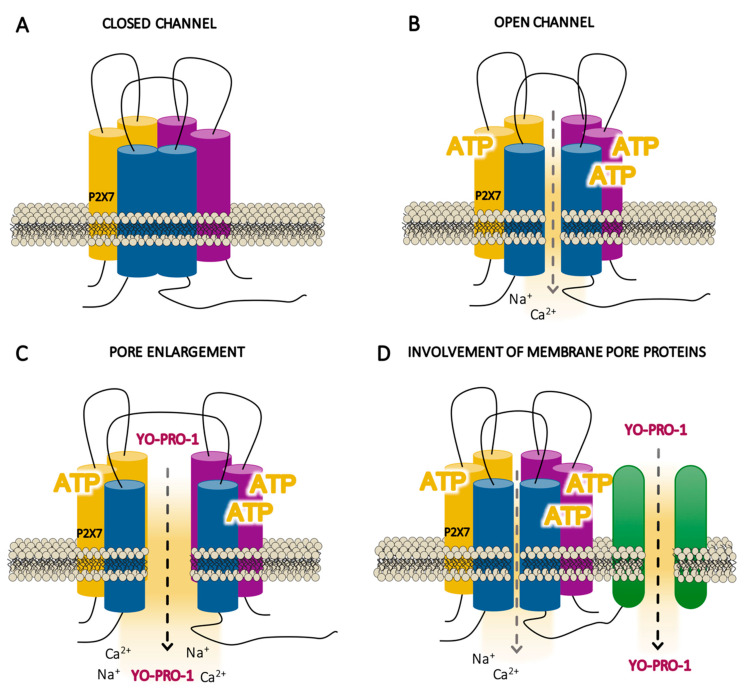
Schematic illustrations of the hypothetical states of the P2X7 receptor (P2X7R) activation. (**A**) Resting state of the P2X7R where the agonist (adenosine triphosphate, ATP) is not present or exists in insufficient concentrations, so that the conformation of the receptor does not open its own ion channel (closed channel). (**B**) Initial activation of the receptor within milliseconds by the three molecules of ATP, inducing a selective channel opening for small cations (open channels). (**C**) This initial state is followed by a prolonged phase within seconds to a minute of enlargement of the receptor itself (pore enlargement), allowing the flux of non-selective large ions such as Yo-Pro-1. (**D**) This prolonged phase of activation can involve other membrane proteins.

**Figure 4 ijms-21-08454-f004:**
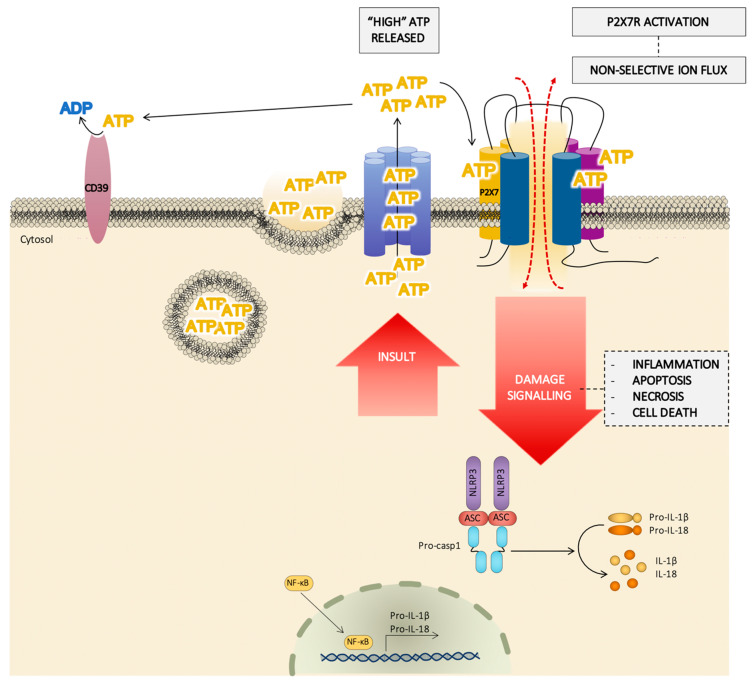
Schematic pathway of the role of the P2X7 receptor (P2X7R) in cell damage signaling. In response to a severe damage or an insult signal, the cell releases a huge amount of adenosine triphosphate (ATP) by different mechanisms (from ATP dense granules, ATP channels, …) and the extracellular concentration of ATP reached could be enough to activate the P2X7R, facilitating a flux of large ions and the activation of the inflammasome. The P2X7R/inflammasome complex induces the synthesis of IL-1β and IL-18, also regulated by NF-κB. This hypothesis agrees with the role of the P2X7R in processes such as inflammation, apoptosis, and necrosis.

**Figure 5 ijms-21-08454-f005:**
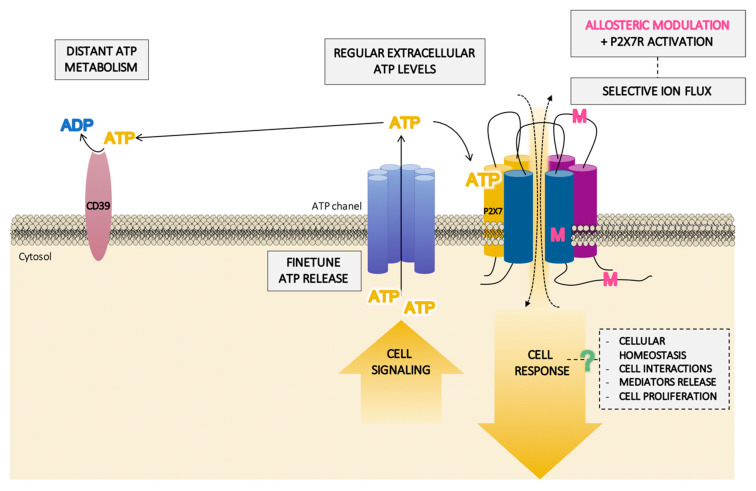
Schematic pathway of the hypothetical role of the P2X7 receptor (P2X7R) in homeostasis balance. In physiological situations, the specific localization of the P2X7R in the membrane, near finetuned adenosine triphosphate (ATP)-released structures and distant from ATP-degrading enzymes, could provide a small extracellular concentration of ATP which are not enough for the activation of the receptor. However, this activation can take place in the presence of a positive allosteric modulator (M) binding to different domains of the receptor. This will facilitate the selective flux of ions through the P2X7R channel, thus inducing a transduction pathway which could lead to a cellular homeostasis response without inducing necessarily cellular damage.

**Table 1 ijms-21-08454-t001:** Endogenous and pharmacological molecules that modulate P2X7 receptor activation in different species and cell types.

**Endogenous Molecules**	**P2X7R** **Modulation**	**Specie/Cell Type**	**References**
Divalent cations Protons	(−)	HEK-293 transfected with rat P2X7R	[136,153]
Phosphoinositides	(+)	HEK transfected Macrophage cell line Primary T cells	[139]
Lysophosphatidyl-choline PLA_2_ lysolipids	(+)	HEK-293 expressing rat, mouse or human recombinant P2X7R	[140]
Glycosaminoglycans	(+)	CHO-K1 cell line	[141]
Lipoglycans *	(+)	Murine RAW 264.7 macrophages	[142]
Nicotinamide adenine dinucleotide (NAD)	(+)	Murine BW5147 T lymphoma Murine bone marrow-derived macrophages HEK 293 transfected with murine P2X7R	[143]
Drug Name	**P2X7R** **Modulation**	**Specie/Cell Type**	**References**
Clemastine	(+) S	HEK-293 transfected with human P2X7R Human-monocyte-derived macrophages Murine bone marrow-derived macrophages	[145]
Tenidap	(+) E/S	J774 mouse macrophage cell line	[35]
Propofol Ketamine	(+)	GMI-R1 rat microglia cell line	[146]
Polymyxin B	(+) E/S	HEK-293 transfected with P2X7R K-562 erythroid transfected with P2X7R J-774 mouse macrophages Human-monocyte-derived macrophages Lymphocytes from lymphocytic leukemia	[36]
Ginsenoside	(+) E/S	J774 mouse macrophage cell line HEK-293 Macrophages from C57BL/6 mice	[37,148,149,154]
Agelasine Garcinolic acid	(+)	HEK-293 transfected with human P2X7R A-375 human melanoma cell line	[150]
Ivermectin	(+) E/S	Human-monocyte-derived macrophages	[151]
GW791343	(+) E/S (−)	HEK-293 transfected with rat P2X7R HEK-293 transfected with human P2X7R	[152]

Increase (+) or reduction (−) in ATP-P2X7R’s actions. For some drugs, the concentration–response curves to the agonist (ATP/BzATP) reflect changes in sensitivity (S) and/or efficacy (E). * endogenous from bacteria origin.

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
