# Peer review of "Structural and Functional Basis for Understanding the Biological Significance of P2X7 Receptor"

_ijms, 2020, doi:10.3390/ijms21228454_

Round 1

Reviewer 1 Report

In this manuscript, Martínez-Cuesta et al. reviewed the structure and function of P2X7 receptor This is a well-conducted and convinced study.They divided the paper into different chapters that provide useful insights for numerous research fields. This is a well-conducted and convinced study. I have only one minor point to address: the authors could address more in details the role of P2X7 receptor in cell proliferation both in wound healing and cancer. The dual role of P2X7 receptor in cell death and cell survival is very intriguing and it would be of high interest to discuss this point in the paper.

Author Response

Referee 1

“In this manuscript, Martínez-Cuesta et al. reviewed the structure and function of P2X7 receptor This is a well-conducted and convinced study. They divided the paper into different chapters that provide useful insights for numerous research fields. This is a well-conducted and convinced study. I have only one minor point to address: the authors could address more in details the role of P2X7 receptor in cell proliferation both in wound healing and cancer. The dual role of P2X7 receptor in cell death and cell survival is very intriguing and it would be of high interest to discuss this point in the paper.

We very much appreciate that referee 1 considers that our manuscript to be a well-conducted and convincing study and that it provides useful insight for numerous research fields. Care has been taken to modify the new version in accordance with his/her suggestions and to discuss the subjects he/she raises more extensively. We believe the comments have improved the manuscript.

Minor Point:

Point 1. ”The authors could address more in details the role of P2X7 receptor in cell proliferation both in wound healing and cancer. The dual role of P2X7 receptor in cell death and cell survival is very intriguing and it would be of high interest to discuss this point in the paper.

Response 1. We have followed referee´s suggestion and we have modified the part of the review entitled ‘Pathophysiological role of P2X7 receptor and its biological significance’ in order to accommodate the role of the receptor in the proliferation in wound healing (Page 13, lines 385-387). The dualism of P2X7 receptor in cell proliferation and cell death in cancer has also been described (Page 13, lines 391-397). The terms cell death and cell proliferation have been included in Figure 4 (Page 9) and 5 (Page10), respectively and references 166, 170-173 have been added to the manuscript (Page 23, lines 890-891 and 901-912).

Reviewer 2 Report

In this review, the authors give a nice overview of P2X7 receptors. Overall, the presentation is fine. However, there are a number of grammatical errrors. These include:

Line 16: "native" should be "natively"

Line 18 (and many others throughout the paper): "Ca+2", should be Ca2+. The authors should go through the entire manuscript to change this since it occurs in other places.

Lines 23-26: The sentence, "Therefore, studies of the human..." is too long and should be split.

Line 34: "recognizes" is odd an odd choice, use another word

Line 35: "excessively" is wrong choice of word, use "high levels"

Line 41: "in" is wrong choice of word

Line 119: "pore formation" is incorrectly used. That is, polymorphisms do NOT increase the formation of pores. They CAN affect the formation, but they don't increase it.

Line 148: "particularly" is the wrong word of choice, pick something else

Line 195: "activation (pore formation)" as used here implies that they are the same processes. They are not the same. So please separate or distinguish both of them.

Line 197: "heterologous" should be heterologously

Line 207: "resting state" is incorrectly used here. It should be closed state---which is what the authors have indicated in the figure. So remove resting state.

Line 221: "undermining" is wrong choice of word here as well.

Line 235: remove the word "exclusively", not appropriate here.

Line 248, 249: change "permeabilization" to permeation. This is the appropriate term

Line 252: "formation" should be forming

Line 264: the title makes no sense. Receptors DO NOT "interact" with pathways. Pathways are processes.

Line 270-274: The sentence "Studies in different cell..." should be split. It is too long.

Line 290: the word "ligated" is incorrectly used here and makes no sense. Use "linked"

Line 331: remove the word "the" from the bacteria

Author Response

Referee 2

In this review, the authors give a nice overview of P2X7 receptors. Overall, the presentation is fine”

We fully appreciate the comments of reviewer 2 and thank her/him for considering that “we give a nice overview of P2X7 receptors and that the presentation is fine”. The manuscript has been modified according to the referee’s suggestions, which we believe have helped to improve our work. The specific responses to the comments of the referee are as follows:

Point 1. Line 16: "native" should be "natively".

Response 1. The term ‘native’ has been replaced by ‘natively’ (Page 1, line 16).

Point 2. Line 18 (and many others throughout the paper): "Ca+2", should be Ca2+. The authors should go through the entire manuscript to change this since it occurs in other places.

Response 2. ‘Ca+2’ has been changed to ‘Ca2+’ throughout the entire manuscript (Page 1, line 18; page 2, line 54; page 2, line 91; page 6, line 213; page 6, line 219 and page 11, line 350).

Point 3. Lines 23-26: The sentence, "Therefore, studies of the human..." is too long and should be split.

Response 3. The sentence has been split into the following two new sentences: “Therefore, studies in human cells that constitutively express P2X7R need to investigate the precise endogenous mediator located nearby the activation/modulation domains of the receptor. Such research could help to understand a possible physiological ATP-mediated P2X7R homeostasis signaling” (Page 1, lines 23-26).

Point 4. Line 34: "recognizes" is odd an odd choice, use another word.

Response 4. ‘Recognizes’ has been changed to ‘utilizes’ (Page 1, line 34).

Point 5. Line 35: "excessively" is wrong choice of word, use "high levels".

Response 5. The word ‘excessively’ has been changed to ‘high levels’ (Page 1, lines 35 and 36).

Point 6. Line 41: "in" is wrong choice of word.

Response 6. The preposition ‘in’ has been changed to ‘under’ (Page 1, line 41).

Point 7. Line 119: "pore formation" is incorrectly used. That is, polymorphisms do NOT increase the formation of pores. They CAN affect the formation, but they don't increase it.

Response 7. The sentence has been modified to include the suggestion of the referee as follows: ‘SNPs have been associated with an increase in ATP-activated ion channel function and affect the formation of pores’ (Page 4, lines 118-120).

Point 8. Line 148: "particularly" is the wrong word of choice, pick something else

Response 8. The word ‘particularly’ has been changed to ‘critical’ (Page 4, line 149).

Point 9. Line 195: "activation (pore formation)" as used here implies that they are the same processes. They are not the same. So please separate or distinguish both of them.

Response 9. We accept the point of the reviewer and agree that activation and pore formation are not the same process. However, we would like to include both concepts in the sentence which now reads as follows: ‘Impaired P2X7R activation and pore formation have been described …..’ (Page 5, lines 195 and 196).

Point 10. Line 197: "heterologous" should be heterologously

Response 10. The word ‘heterologous’ has been modified to ‘heterologously expressed’, according to referee’s suggestions (Page 6, line 197).

Point 11. Line 207: "resting state" is incorrectly used here. It should be closed state---which is what the authors have indicated in the figure. So remove resting state.

Response 11. According to the referee’s suggestion, the term ‘resting state’ has been changed to ‘closed state’ (Page 6, line 207).

Point 12. Line 221: "undermining" is wrong choice of word here as well.

Response 12. The term undermining has been changed to reduction (Page 6, line 221).

Point 13. Line 235: remove the word "exclusively", not appropriate here.

Response 13. The word exclusively has been removed and the sentence has been modified as follows ‘P2X7R are not the only receptors expressed by native cells’ (Page 6, line 235).

Point 14. Line 248, 249: change "permeabilization" to permeation. This is the appropriate term.

Response 14. According to the referee’s suggestion, the term ‘permeabilization’ has been changed to ‘permeation’ (Page 7, lines 248 and 249).

Point 15. Line 252: "formation" should be forming.

Response 15. The word ‘formation’ has been changed to ‘forming’ (Page 7, line 252).

Point 16. Line 264: the title makes no sense. Receptors DO NOT "interact" with pathways. Pathways are processes.

Response 16. We agree with the referee that receptors do not interact with intracellular signaling pathways. The word ‘interact’ has been changed to ‘trigger’ (Page 8, line 264).

Point 17. Line 270-274: The sentence "Studies in different cell..." should be split. It is too long.

Response 17. The sentence has been split into two and now reads as follows: ‘Studies in different cells – for example in rat microglia - suggest that the P2X7R is coupled to c-Jun N-terminal kinase (JNK) and p38, but not to ERK. This association is responsible for the activation of JNK and p38 via a protein tyrosine kinase-dependent mechanism that involves the release of tumor necrosis factor (TNF-α) (Page 8, lines 270-274).

Point 18. Line 290: the word "ligated" is incorrectly used here and makes no sense. Use "linked".

Response 18. The term ‘ligated’ has been changed to ‘linked’ (Page 8, line 290).

Point 19. Line 331: remove the word "the" from the bacteria

Response 19. The word ‘the’ from the bacteria has been removed (Page 10, line 332).